# A Group Intervention to Promote Resilience in Nursing Professionals: A Randomised Controlled Trial

**DOI:** 10.3390/ijerph19020649

**Published:** 2022-01-06

**Authors:** Gesche Janzarik, Daniel Wollschläger, Michèle Wessa, Klaus Lieb

**Affiliations:** 1Leibniz Institute for Resilience Research (LIR) Mainz, 55122 Mainz, Germany; wessa@uni-mainz.de (M.W.); klaus.lieb@unimedizin-mainz.de (K.L.); 2Institute of Medical Biostatistics, Epidemiology and Informatics (IMBEI), University Medical Center of the Johannes Gutenberg University Mainz, 55131 Mainz, Germany; wollschlaeger@uni-mainz.de; 3Department of Clinical Psychology and Neuropsychology, Johannes Gutenberg University Mainz, 55122 Mainz, Germany; 4Department of Psychiatry and Psychotherapy, University Medical Center of the Johannes Gutenberg University Mainz, 55131 Mainz, Germany

**Keywords:** mental health, resilience, nursing, occupational stress, psychotherapy, coping, randomised controlled trial

## Abstract

In this study, a new group intervention program to foster resilience in nursing professionals was tested for efficacy. In total, 72 nurses were recruited and randomised to either an intervention condition or to a wait list control condition. The study had a pre-test, post-test, follow-up design. The eight-week program targeted six resilience factors: cognitive flexibility, coping, self-efficacy, self-esteem, self-care, and mindfulness. Compared to the control group, the intervention group reported a significant improvement in the primary outcome mental health (measured with the General Health Questionnaire) from pre-test (*M* = 20.79; *SD* = 9.85) to post-test (*M* = 15.81; *SD* = 7.13) with an estimated medium effect size (*p* = 0.03, *η^2^* = 0.08) at post-test. Further significant improvements were found for resilience and other resilience related outcomes measures. The individual stressor load of the subjects was queried retrospectively in each measurement. Stress levels had a significant influence on mental health. The intervention effect was evident even though the stress level in both groups did not change significantly between the measurements. Follow-up data suggest that the effects were sustained for up to six months after intervention. The resilience intervention reduced mental burden in nurses and also positively affected several additional psychological outcomes.

## 1. Introduction

The work of nurses in hospitals is associated with a high psychological and physical workload. Nurses work in a complex, constantly changing and potentially adverse work environment. The diverse physically and emotionally demanding activities often conflict with a lack of personal autonomy, low external support, and limited resources [1]. The global problem of increasing stress and associated problems at the employee, patient, and organisational level is often discussed in the international nursing literature. This critical development may be further intensified by structural changes and staff shortages in nursing care [2]. High work demands may have negative effects on nurses’ work performance, health and wellbeing [3]. Recent studies indicate that a high number of nurses intend to leave the profession due to high job stress and dissatisfaction [4]. Those who stay are at risk of burnout [5]. High stress is a risk factor for problematic substance use among nurses [6]. Moreover, shift work [7] and bullying in the workplace [8] may have a negative impact on mental health. Nurses with mental health problems in turn may contribute to a poorer quality of care and reduced patient safety [9]. One starting point to reduce the negative effects of high stress can be the implementation of training programs that foster resilience and support nurses to better cope with their demands of their clinical work [10].

Resilience can be understood as a multidimensional and dynamic process of positive adaptation to stress [11,12], which involves a complex interaction between individual traits and the environment the individual is living in [13,14]. The result of this process is characterised by the maintenance or quick recovery of mental health during or after stress [11,12,15]. Such mental health outcome is associated with a variety of resilience factors, e.g., of psychological, biological, neurocognitive, genetic, or social resources [14], which interact with one another (e.g., [16]). As resilience is a dynamic process, it can be subject to change during, e.g., periods of stress, and this is true in both directions. Therefore, resilience can be trained [17], and psychological interventions to foster resilience in healthcare professionals and students have indeed the potential to improve mental health [18,19,20].

Recent systematic reviews have indicated that intervention studies often show a number of weaknesses regarding the methodological quality and the operationalisation of resilience, e.g., low sample size, absence of random sampling, divergent efficacy measures to assess resilience, no or only short-term follow-up measurements, and a lack of relating the mental health status to the respective level of experienced stress [18,19,21].

The present study aimed to improve upon the methodological quality of previous intervention studies, specifically by also examining the influence of stress levels on mental health and intervention effects. The systematic reviews mentioned above show that the training contents of resilience interventions are heterogeneous and refer to different theoretical foundations. The contents are commonly based on approaches of cognitive behaviour therapy, (CBT) or combine CBT with other training methods, e.g., with acceptance and commitment therapy (e.g., [22]). There are also several interventions that are only based on one theoretical foundation, e.g., on mindfulness-based stress reduction (e.g., [23]). However, the conceptual approach of integrating psychodynamic with CBT elements seems to be a novelty in resilience interventions research. Since caregivers in hospital are constantly confronted with complex, emotionally, and socially demanding situations, the use of psychodynamic approaches can be particularly helpful for this professional group. As part of the present training, the nurses, therefore, also learn how their biographical background influences their emotions, cognitions, and behaviours of today. They obtained trained to identify their maladaptive defence mechanisms, which, in turn, are regarded as the main cause of being stressed. They learn that they use those defences in order to avoid difficult emotions. Accordingly, the main focus of the training is the improvement of emotion regulation. According to their individual conflicts, the nurses can also choose between various coping strategies that are based on different therapeutic techniques, e.g., on CBT, mentalisation-based treatment, or on mindfulness-based stress reduction. The group intervention (“*The new Growth*”) evaluated for this study was designed to promote nurses’ mental health and resilience by helping them to better cope with their work-related and private stressors. Psychotherapy research shows that, depending on the patient and the context, the integration of therapeutic approaches from different schools can effectively contribute to the success of the treatment, and is more flexible to patients’ needs and allows a better adaptation of the therapy [24]. The studied intervention allows a certain flexibility in the implementation. The nurses can set their own priorities and have free choices between exercises from different therapeutic approaches. The program targeted six psychological resilience factors identified from empirical evidence, which are shortly introduced in the following paragraphs.

Coping encompasses all cognitive and behavioural efforts a person undertakes to manage his or her stressful demands that tax or exceed his or her personal resources. Depending on the characteristics of the situation, either problem-focused or emotion-focused coping strategies can be useful. When using problem-focused strategies, a person is acting on the environment or oneself in order to change the stressful situation. Emotion-focused strategies can help a person to adapt to the stressor by emotion regulation, for example, by changing the relational meaning of a stressful event. Especially in situations in which nothing useful can be done to change the problem, problem-focused strategies can be counterproductive and emotion-focused efforts would be the a better choice [25]. The acquisition of effective coping strategies is of high importance particularly for nursing professionals [26]. Studies indicate that the use of active coping strategies can improve the resilience of nurses [27]. Psychological interventions promoting coping strategies in nurses can reduce burnout [28]. The present intervention involves both the training of problem-focused strategies to enhance active coping behaviour and emotion-focused strategies to promote emotion regulation for example by practicing positive reframing.

Cognitive flexibility has many facets, including the ability to change perspectives (e.g., to view a situation from different perspectives or to adopt perspectives from others], to quickly switch between tasks, to approach problems flexibly, or to adjust to changed demands or priorities [29]. Within the *Positive Appraisal Style Theory* of Resilience, cognitive flexibility is conceptualised as a general resilience mechanism, as it is supposed to allow individuals to flexibly regulate emotional responses to potential stressors, thereby protecting against the harmful effects of stress [12]. Furthermore, higher psychological flexibility has been proven to be associated with better mental health, better physical health, better job performance, and less stress [30]. Thus, the improvement of cognitive flexibility has been chosen as one core element of various psychotherapeutic approaches, e.g., cognitive restructuring [31], mentalisation-based treatment [32], or acceptance and commitment therapy [33].

Self-efficacy is the belief in one’s competence to tackle difficult or novel tasks and to cope with adversity in specific demanding situations. People with a high self-efficacy trust in their own abilities, choose more challenging tasks, and have ambitious goals and stick to them. They motivate themselves and recover more quickly from setbacks. They experience a low level of negative emotions in a threatening situation [34]. It is one of the resilience factors most robustly and closely related to resilience [35]. High self-efficacy has a positive impact on the resilience of caregivers [27] and has been associated with well-being [36], positive affect, and work performance [37] in nurses. Furthermore, a negative relationship of self-efficacy with burnout has been observed [38].

Self-esteem is defined by how much value people place on themselves and is the evaluative component of self-knowledge [39]. Studies imply that the factor, self-esteem, is strongly associated with resilience and self-efficacy [37,40]. Some studies support the hypothesis that self-esteem buffers against stress and traumatic experiences (e.g., [41]). In addition, studies have shown positive associations with self-compassion [42], well-being [43], hope, and optimism [41]. Negative associations were found with depression, anxiety and, stress (e.g., [42]).

Self-care can be defined as “those activities individuals undertake in promoting their own health, preventing their own disease, limiting their own illness, and restoring their own health” ([44], pp. 181). Self-care includes various behaviours, that balance the effects of emotional and physical stress, e.g., healthy eating, getting enough sleep, yoga, meditation, mindfulness, relaxation, building meaningful relationships with others, etc. [45]. Self-care is essential for the nursing profession by having a positive impact on nurses’ wellbeing and patient care [45,46]. However, there are numerous barriers in the workplace of nurses that hinder participation in health-promoting self-care [47]. A deficit in self-care is associated with higher levels of burnout [48]. A number of studies demonstrate the positive effects of self-care interventions in increased mental health of caregivers(e.g., [49]).

Mindfulness can be defined as “a way of relating to oneself and the world that is characterised by curiosity, openness, and acceptance” ([50], pp. 31). Mindfulness meditation causes neuroplastic changes in brain regions that are involved in the processes of attention control, emotion regulation, and self-perception [51]. Several nursing studies show positive correlations between mindfulness and resilience (e.g., [52]). A meta-analysis investigating the effect of mindfulness trainings (interventions between three and twelve weeks) indicates, that exercises like meditation, body scan, yoga, and breathing exercises can reduce burnout in nurses, regardless of the duration of the interventions [53].

It is hypothesised that participation in the training leads to a reduction in mental health problems depending on the individual level of experienced stress. It is expected that an increasing number of experienced stressors within a week is associated with a poorer mental health of the nurses. It is also assumed that participation in the training leads to an enhancement of resilience, wellbeing, satisfaction with life, self-esteem, self-efficacy, emotion regulation, and coping, as well as to a reduction in perceived stress. It is hypothesised that the differences between intervention and control group will sustain for the follow-up period.

## 2. Materials and Methods

This randomised controlled trial included three assessment points: pre-test, post-test, and three follow-up measurements at three, six, and nine months.

Subjects were randomly assigned to either an intervention or a waiting list control group. In order to consider the various working conditions in different medical departments (e.g., neurology, surgery, dermatology, etc., stratified randomisation with the medical department as stratification factor was used.

The intervention was an eight-week group psychotherapy intervention to foster resilience. The pre-test (t0) was conducted one week before the first training session, the post-test (t1) within one week after the last training session. Follow-up measurements were conducted three (t2), six (t3), and nine months (t4) after intervention. No blinding was involved in the study. Due to the beginning coronavirus pandemic, the control group could not take part in the last measurement and received the intervention after t3. The study was approved by the Ethics Committee of the Medical Association of Rhineland-Palatinate (application number: 2019-14276). At the commencement of the study, all participants provided written informed consent. The CONSORT checklist provides information on fulfilment of CONSORT requirements [54] (see Appendix A).

### 2.1. Procedure and Recruitment

Recruitment and intervention took place at the Medical Centre of the Johannes Gutenberg-University Mainz (Germany). To the time of the study, the University Medical Centre had approximately 8500 employees including about 4000 nurses, and approx. 1600 inpatient beds. The director of nursing was contacted to initiate the collaboration. After approval by the administration, responsible nursing managers informed the nursing staff. Nurses from direct patient care and nurses in leadership positions were recruited for the study. All interested employees were sent written information about the study. Registered nurses received a two-hour introductory session for detailed information, completed by a screening to examine inclusion and exclusion criteria. The structured and fully standardised clinical interview for mental health (M.I.N.I., [55]). was used for comprehensive diagnostic assessment. Signing the informed consent and completing the baseline questionnaire completed the enrolment process. The following inclusion/exclusion criteria were applied: participants had to be healthy, be at least 18 years old and had to have appropriate German language skills. They were not allowed to be on any psychopharmacological medication or had to keep the medication on a stable level during the study. Subjects were excluded if they had any psychotherapy treatment during the study, had a severe mental illness, suicidal tendencies, substance abuse, serious physical illnesses, or other physical or mental restrictions, which may hinder participation (e.g., impaired hearing). Sample size planning for a covariance analysis (ANCOVA) was conducted with G*Power version 3.1.9.4 [56]. A sample size of 60 was needed to obtain 0.60 power assuming an estimated correlation between pre- and post-test of 0.5, a medium effect size of f = 0.29 and statistical significance level = 0.05. Recruitment started in February 2019; training implementation began in June 2019. Nurses participated voluntarily and did not receive any payment.

### 2.2. Outcome Measures

Primary outcome measure was a self-reported indicator of mental health problems assessed by the General Health Questionnaire-28 (GHQ-28; [57]), a widely applied screening instrument to detect emotional distress and possible psychiatric symptoms during the last weeks. The 28 items are rated on a 4-point Likert-type scoring from 0 (= not at all) to 3 (= much more than usual). The total score ranged from 0 to 84. A total score of 23 or below can be classified as a non-psychiatric case, while a total score ≥24 may be classified as a probable psychiatric case, but this score is not an absolute cut-off [58]. In numerous studies, the psychometric indices were rated as satisfactory (e.g., [59]). In the present sample, the Cronbach’s α for the total scale was 0.91. To investigate the impact of stress on mental health, the Mainz Inventory of Microstressors (MIMI; [60]) was used. The MIMI allows the retrospective assessment of microstressors over a one-week period. The questionnaire comprises 58 items, which inquire about the occurrence of micro-stressors during the last seven days. Moreover, it separates the stressor occurrence from perceived stressor severity. For evaluating the intervention, merely the stressor occurrence was analysed.

Secondary outcome measures included self-reported indicators of resilience assessed with the Brief Resilience Scale (BRS; [61,62,63]) and the Connor–Davidson resilience scale (CD-RISC; [64]). While the BRS conceptualizes resilience as the ability to recover from stress, the CD-RISC operationalises resilience as a composite of resilience factors. Additionally, several other validated instruments were used to measure resilience related constructs such as subjective wellbeing (WHO-5; [65]), general self-efficacy (SWE; [66]), self-esteem (RSES; [67]), satisfaction with life (SWLS; [68]), emotion regulation skills (ERSQ-27; [69]), perceived stress (PSS; [70,71]). In order to assess problem-focused und emotional-focused coping behaviour the two scales active coping and positive reframing of the Brief-COPE [72] were used.

### 2.3. Intervention

The intervention comprised eight weekly sessions of two hours each held in a group setting with five to 10 participants per group. G.J., a psychologist in post-graduate education as a psychological psychotherapist, conducted all training sessions. The training included therapy elements from cognitive behavioural therapy and psychodynamic psychotherapy. Additionally, mindfulness and imagination exercises were included. The aim of the intervention was to provide participants with new skills to help them cope better with individual stressors. The training was based on a psychodynamic model focusing on stress development. In the first two training sessions, the nurses’ private and occupational problems were analysed and an initial psychodynamic understanding of the stressors was developed. The nurses considered how the experience of stress relates to difficult relationship experiences in their childhood. Building on this, they learned which inner psychic conflicts are the triggers for their stress. They also identified frequently used dysfunctional defence mechanisms (e.g., rationalisation, regression, reaction formation, isolation of affect, autoaggression, etc.). This psychodynamic model formed an important basis for the rest of the training and was always an indirect part of the other sessions as well. For the development of sessions three to five, six resilience factors were initially selected based on their empirical evidence: cognitive flexibility, coping, self-efficacy, self-esteem, self-care, and mindfulness. In order to improve these factors, exercises based on various therapeutic techniques were developed, namely, mentalization exercises and exercises about cognitive restructuring to train cognitive flexibility, progressive muscle relaxation (PMR) and mindfulness exercises to improve self-care or mindfulness, gathering strengths and skills to improve self-efficacy, etc. Training sessions six and seven comprised a problem-solving process in which the nurses focused on one of their main problems from the beginning of the training. In order to solve this problem, the nurses selected their appropriate strategies and skills, transferred them to everyday life and tested them for their usefulness. The last training session included reflections on personal development over the past eight weeks. There were exercises for a more realistic and positive self-image in which the participants gave each other feedback. In addition, discussions about one’s own values in life and the goals after the training were initiated. Exercises were performed in the form of individual, pair, and group work. Homework assignments were incorporated to consolidate the subject matter. The training was tested for feasibility in a preliminary study and then slightly modified again. Table A1 shows an overview of the training content.

### 2.4. Statistical Analysis

Descriptive statistics including means, standard deviations, and frequency distributions were calculated for initial analyses. To examine possible differences between intervention and control group, χ^2^-Test, Wilcoxon-Test, and exact Fisher test were used. Q-Q plots for both groups were used to detect deviations from normal distribution. The intervention effect was tested with an analysis of covariance (ANCOVA) with the post-test score as dependent variable, the pre-test score as the covariate and the treatment allocation as a fixed factor. The stressor load, i.e., the number of daily hassles was included as an additional covariate to examine the influence of stress on mental health. The analysis was based on the intention-to-treat principle. Subjects with missing values were excluded from the analysis using ANCOVA, but not from linear mixed models. Missing data were not imputed.

To estimate the intervention effect over time (t1–t4) on different outcomes, linear mixed models (LMM] with random intercepts for participant were used as implemented in package lme4 [73] for the statistical environment R [74]. Treatment was the fixed effect of interest, and the outcome at t0 as well as time were continuous covariates. Due to the small sample size, group-by-time interactions could not be analysed. *p*-values for coefficients in LMM were calculated based on t-tests with Satterthwaite’s correction to the denominator degrees of freedom as implemented in package lmerTest [75]. *p*-values for the main outcomes of interest were validated in model comparisons of respective full models against restricted models using Kenward–Roger corrected denominator degrees of freedom as implemented in package pbkrtest [76]. Since both methods always agreed up to two significant digits, only *p*-values based on Satterthwaite’s method are reported here. In diagrams, nonparametric trend lines were fitted using locally weighted linear regression (loess). All statistical tests were two-tailed; the significance level was set at ɑ = 0.05. The effect size was estimated using partial eta squared (*η*^2^). According to [77], *η^2^* = 0.01 is considered as a small effect, *η*^2^ = 0.06 as a medium effect, and *η*^2^ = 0.14 as a large effect. R version 4.0.2 was used for statistical analysis.

## 3. Results

### 3.1. Participants

As shown in Figure 1, 72 nurses were enrolled and met eligibility criteria. Because some nurses signed-off before enrolment, three subjects could not be assigned randomly in order to ensure the same composition of both groups. The intervention group comprised 38 subjects, the control group 34.

Socio-demographic data are presented in Table 1. There were no significant differences between intervention and control group regarding age, gender, marital status, weekly working hours, and stressor load before the intervention. With regard to the entire nursing staff of the clinic, the representativeness of the sample can be assumed, since nurses from many different medical departments participated in the study.

### 3.2. Primary Outcome

As shown in Table 2, the ANCOVA for the primary outcome (GHQ-28 sum score) revealed a statistically significant difference between the intervention and the control group from pre-test to post-test (*t* = −2.25, *p* = 0.03, *η*^2^ = 0.08). As expected, there was a statistically significant association of the stressor load with mental health at post-test (*t* = 3.64, *p* ≤ 0.001, *η*^2^ = 0.19). Figure A1 illustrates that the number of mental health problems increased with the growing number of daily hassles. When controlling for the stressor load, the intervention effect changed only marginally (*t* = −2.28, *p* = 0.03, *η*^2^ = 0.08).

Descriptive statistics for the follow-up measurements (t2–t4) are presented in Table 3. The results of the mixed model analysis (Table 4) indicated also for the follow-up period a significant group effect for mental health (*β* = −4.18, *t* = −2.67, *p* = 0.01, *η*^2^ = 0.11). Separated nonparametric trend lines for both groups are illustrated in Figure A2. The association of stressor load with mental health was statistically significant for the follow-up period as well (*β* = 0.10, *t* = 5.29, *p* ≤.001, *η*^2^ = 0.13). The intervention effect was evident even though the stress level in both groups did not significantly change between the measurements (Figure A3).

### 3.3. Secondary Outcomes

The analysis of the treatment outcome using ANCOVA indicated further statistically significant group effects from pre-test to post-test for resilience measured with the CD-RISC, active coping, satisfaction with life, and emotion regulation skills. No statistically significant group differences from pre-test to post-test were found for the secondary outcomes’ wellbeing, resilience measured with the BRS, perceived stress, self-efficacy, self-esteem, and positive reframing (Table 2). The mixed model analysis revealed statistically significant group effects for the follow-up period for resilience measured with the CD-RISC, wellbeing, emotion regulation skills, active coping, and positive reframing. No statistically significant group differences for the follow-up period could be found for the outcomes’ resilience measured with the BRS, satisfaction with life, perceived stress, self-efficacy, and self-esteem. An overview of all group effects for the follow-up period is shown in Table 4.

## 4. Discussion

The results of this RCT confirm the positive effect of the newly developed resilience training on the primary outcome mental health. The intervention was also shown to be effective in improving resilience, wellbeing, emotion regulation, and coping in nurses. Compared to the control group, the number of mental health problems in the intervention group was significantly lower after the intervention. Medium effect sizes were found at post-test for mental health. These positive effects sustained for up to six months after the intervention. At the nine-month follow-up, however, mental health scores fell back to nearly the level of the baseline measurement. As already shown by previous intervention studies, the training of specific resilience factors can improve mental health (e.g., [78]). In the meta-analysis about resilience interventions for health professionals of [19], the subgroup analysis for group interventions revealed a standardised mean difference (SMD) of −0.41 (−0.69, −0.13) for the primary outcome depression. In comparison, the present training was found to be somewhat more effective in improving mental health. Interpreting the results, it should be noted that 25 subjects (34.7% of the total sample) had a total GHQ-score ≥24, which may explain the better results at least in part. As proposed by [58], a cut-off score above 24 indicates a possible mental disorder, which should be further assessed. The findings also show that the stressor exposure as measured by the Mainz Inventory of Microstressors (MIMI; [60]) had a large effect on mental health. By controlling for the individual stressor load, however, we were able to show that mental health problems were reduced due to the intervention and not due to a decreasing level of stress. It can be assumed that by participating in the training, the nurses learned new ways to better manage their work stress and thus reduce the negative effects of stress on mental health. So far, only a few studies on resilience interventions controlled for individual stressor load (e.g., [79]).

The results also show a positive change in resilience measured with the CD-RISC. The findings are comparable to the effect found in other randomised intervention studies (e.g., [22,80]). The effect also sustained up to the six-month follow-up. The results of the CD-RISC are not entirely consistent with those of the BRS. No significant group differences could be found in resilience measured with the BRS. As mentioned above, the two questionnaires assess different aspects of resilience and there is no ‘gold standard’ to quantify resilience [21]. The CD-RISC focuses on assessing the availability of different resilience factors (e.g., positive acceptance of change, and secure relationships), whereas the BRS measures resilience as an outcome, thus as the ability to recover from stress. While the CD-RISC is a frequently used instrument to evaluate the efficacy of resilience trainings (e.g., [81,82]) and is considered to be sufficiently sensitive to change [64], we could not find any studies that examined the sensitivity to change for the BRS. It is possible that minor changes in resilience over time cannot be adequately assessed with the BRS. The understanding of resilience as a multidimensional, dynamically changeable adaptation process [12,14,15] also raises the fundamental question whether a valid measurement of resilience is only useful by measuring the reactivity of individuals’ mental health to stressors during an interval of several weeks or months (see [83]).

No significant group effects in perceived stress as measured by the PSS [70] were observed. Previous intervention studies reported a significant reduction in stress with medium to large effects sizes from pre-test to post-test (e.g., [22,82]). The present training thus proved to be less effective for stress prevention than other comparable interventions. This could be related to the fact that the intervention did not specifically focus on reducing stress in particular, but rather on understanding and overcoming individual problems in daily life and at working on inner conflicts.

While no significant group differences for self-esteem and self-efficacy could be found, a short-term effect from pre-test to post-test for satisfaction with life was detected. Further long-term effects were observed for wellbeing, active coping, positive reframing, and emotion regulation. During the training, the nurses actively cope with their everyday problems. Active dealing with fundamental life issues appeared to lead to higher satisfaction with life and better coping behaviour immediately after intervention. The training of cognitive and socio-emotional skills (e.g., cognitive and emotional reframing difficult situations through mentalisation techniques) probably enabled the nurses to maintain their positive emotions over a longer period of time and helped them feel well in the months after training. The effects of these outcomes were sustained for about half a year. By improving awareness, perception, and understanding of emotional reactions, the improvement of emotion regulation skills was one of the central training components successes. In terms of improving emotional competencies, the present intervention is similarly effective to the resilience intervention for nurses, examined in the study of [22]. In this study, the intervention was based on mindfulness and acceptance training, cognitive behavioural training, and solution focused group work.

### Limitations

The study had some limitations. Seventy-two nurses were recruited and randomised to either an intervention condition or to a wait list control condition. The available sample size restricted the complexity of the model, and implied low power especially for testing interaction effects. The calculated sample size was based on the target of achieving a test power of 60%. Recruiting for intervention studies generally involves a number of challenges [84]. The complex bureaucratic and organisational conditions of the hospital made the recruitment process even more difficult. Due to understaffing, changing shift work and high workload, many interested nurses were unable to take part in the time-consuming intervention. These problematic circumstances are likely to occur in most major hospitals. This illustrates the importance of detailed recruitment planning for intervention studies with healthcare professionals in hospitals. Participant attrition during the study highlights the need for the occupational environment to proactively accommodate a time-consuming intervention. Organisational conditions should support interested healthcare professionals to participate in all sessions to ensure the intervention can be evaluated. Another methodological weakness was the lack of an active control group. Since the present intervention comprised several different psychotherapeutic approaches, it was not possible to develop an equivalent control intervention in which the content did not partially overlap with the content of the resilience training.

A further limitation results in the reliance on self-assessment questionnaires. The use of more objective methods, like neuropsychological tests for assessing cognitive flexibility and emotion regulation could be considered. Moreover, the analysis of biomarkers (e.g., cortisol level] could be integrated and has already been applied in previous intervention studies to promote resilience in health professions (e.g., [85,86]). Although the sample consisted of nurses from various medical departments, it cannot be clearly determined whether the results are representative of other hospitals or other cultural circumstances. Some potential influencing work-related variables (e.g., salary and shift work), as well as therapists and group characteristics (e.g., group cohesion) were not collected in the present study. Further studies should examine the influence of these factors on therapy outcome. A multicentre study in several hospitals could be considered in order to achieve a larger sample size and reduce clinic-specific influencing factors. This would further improve the representativeness of the sample and the generalisability of the results. Since a comparison with the control group at t4 was not possible, it remains unclear whether the intervention still affects mental health after the six-month follow-up. According to [19], the positive effects of resilience interventions for health professionals are mostly no longer evident after six months.

## 5. Conclusions

Improving resilience of nurses in hospitals is of great importance for employees, employers and patients alike. In a randomised control trial, the newly developed resilience training, The New Growth, was tested for efficacy. The integration of psychodynamic and behavioural therapy approaches seems to be a novelty in resilience intervention research for nurses. Future studies, however, would benefit from examining which training components turn out to be particularly helpful for promoting resilience. The intervention was successfully implemented and had a positive impact on mental health. The organisational impact of the intervention could be investigated in a further study. Influencing mediator and moderator variables should be determined and the effects on job satisfaction and other work-related factors should be examined. This is particularly important because a firm implementation of the intervention in a hospital would involve financial and time investments. Nevertheless, the positive results suggest that a long-term implementation could potentially lead to higher job satisfaction, less fluctuation, and improved quality of care, which could be convincing facts for employers and insurance companies to invest. Our results also show that determining the efficacy of a resilience intervention, the individual stressor load should always be considered. Although our results indicate that the positive effects on mental health and resilience were sustained for up to six months after the intervention, further intervention studies should examine whether booster sessions can lengthen the long-term effects. However, many interested nurses could not take part in the time-consuming study because of understaffing, shift work, and high workload. These circumstances illustrate that the promotion of individual resilience is only one element to support nurses to overcome high workplace stress. Structural changes and environmental preventions in the hospital, which relate to the creation of better working conditions, are equally essential.

## Figures and Tables

**Figure 1 ijerph-19-00649-f001:**
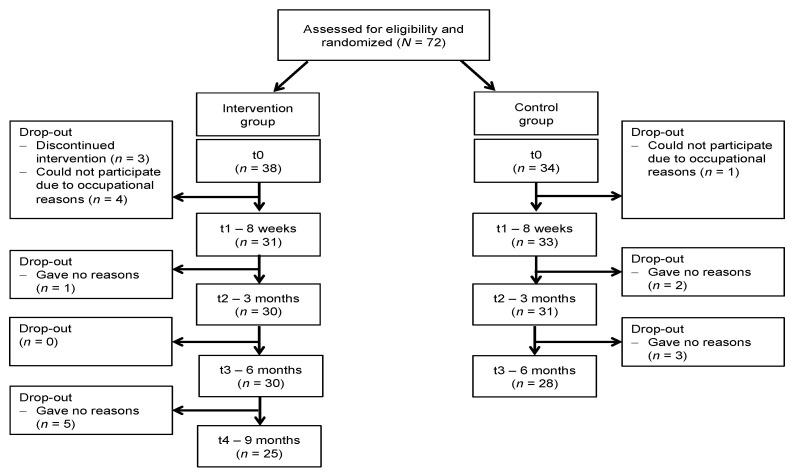
Flowchart of the study.

**Table 1 ijerph-19-00649-t001:** Socio-demographic characteristics of the sample at baseline, t0 (N = 72).

Characteristics	Intervention Group (*n* = 38)	Control Group (*n* = 34)	Test-Statistic	*p*
age in years, M (SD)	47.4 (10.8)	46.5 (10.4)	618 ^1^	0.76
gender, *n* (%)				0.89 ^3^
female	35 (92.1)	31 (91.2)		
male	3 (7.9)	3 (8.8)		
marital status, *n* (%)			4 ^2^	0.17
single	11 (29.0)	17 (50.0)		
married	19 (50.0)	13 (38.2)		
divorced	5 (13.2)	3 (8.8)		
living separately	1 (2.6)	1 (2.9)		
widowed	2 (5.3)	0 (0.0)		
weekly working hours, M (SD)	34.3 (7.6)	33.4 (7.9)	546 ^1^	0.24
Stressorload ^4^, M (SD)	60.82 (24.63)	66.29 (34.07)	693 ^1^	0.60
medical department, *n* (%)				
anaesthesia, intensive care, surgery, emergency medicine, cardiology	11 (30.0)	12 (35.3)		
ophthalmology	3 (7.9)	4 (11.8)		
dermatology	1 (2.6)	1 (2.9)		
gynaecology, urology	3 (7.9)	1 (2.9)		
oral and maxillofacial surgery, ear, nose, and throat medicine	4 (10.5)	3 (8.8)		
neurology, stroke unit	8 (21.1)	8 (23.5)		
nuclear medicine	1 (2.6)	1 (2.9)		
paediatrics	2 (5.3)	0 (0.0)		
psychiatry, psychosomatic medicine	2 (5.3)	1 (2.9)		
OR management	3 (7.9)	3 (8.8)		

^1^ = test-statistic W of the Wilcoxon-test, ^2^ = χ^2^-value, ^3^ = *p*-value for Fisher’s exact test, ^4^ = measured with the Mainz Inventory of Microstressors (MIMI).

**Table 2 ijerph-19-00649-t002:** Descriptive statistics for pre-post data and analysis of group differences using ANCOVA for outcomes at t1 adjusted for the outcome at t0.

	Intervention Group				Control Group						
	t0 (*n* = 38)		t1 (*n* = 31)		t0 (*n* = 34)		t1 (*n* = 33)				
Outcome Measure	*M*	*SD*	*M*	*SD*	*M*	*SD*	*M*	*SD*	*t*	*p*	*η* * ^2^ *
mental health (GHQ-28)	20.79	9.85	15.81	7.13	20.68	8.48	20.03	10.69	−2.25	0.03	0.08
wellbeing (WHO-5)	15.90	4.83	17.26	4.17	15.47	5.51	15.24	5.89	1.59	0.12	0.04
resilience (BRS)	3.08	0.34	3.12	0.32	2.89	0.36	2.85	0.50	1.43	0.16	0.03
satisfaction with life (SWLS)	24.21	5.28	26.71	4.49	24.94	5.06	25.15	4.90	2.05	0.05	0.07
perceived stress (PSS-10)	21.91	3.68	20.03	2.94	19.34	3.37	20.06	3.34	−1.52	0.13	0.04
self-esteem (RSES)	35.37	4.09	35.23	4.52	34.06	3.94	34.36	4.44	−0.17	0.87	0.00
self-efficacy (SWE)	29.95	3.48	30.71	4.02	30.03	3.91	29.73	4.02	1.93	0.06	0.06
emotion regulation (SEK-27)	78.16	12.22	83.55	13.42	74.86	12.83	72.70	14.80	3.13	0.003	0.14
resilience (CD-RISC)	70.40	11.84	73.36	12.38	69.88	11.85	69.33	12.35	2.36	0.02	0.08
active coping (Brief-COPE)	5.18	1.47	5.84	1.46	5.35	1.37	4.94	1.46	2.94	0.01	0.12
positive reframing (Brief-COPE)	5.19	1.60	5.48	1.63	5.09	1.48	5.18	1.42	1.24	0.22	0.02
stressor load (MIMI; occurrence)	60.82	24.63	60.26	26.68	66.29	34.07	63.45	39.2			

**Table 3 ijerph-19-00649-t003:** Descriptive statistics for follow-up data (t2–t4).

	Intervention Group						Control Group			
	t2 (*n* = 30)		t3 (*n* = 30)		t4 (*n* = 25)		t2 (*n* = 31)		t3 (*n* = 28)	
Outcome Measure	*M*	*SD*	*M*	*SD*	*M*	*SD*	*M*	*SD*	*M*	*SD*
mental health (GHQ-28)	18.53	8.11	16.37	7.62	21.16	7.37	23.58	11.20	22.21	10.30
wellbeing (WHO-5)	17.70	4.42	18.10	4.21	14.32	5.17	14.19	5.20	13.60	5.34
resilience (BRS)	3.73	0.65	3.80	0.58	3.79	0.59	3.45	0.83	3.57	0.78
satisfaction with life (SWLS)	26.50	4.10	27.37	4.76	26.76	4.88	25.94	4.84	25.61	5.64
perceived stress (PSS-10)	19.63	3.42	19.63	3.01	20.44	3.36	20.00	3.29	18.93	3.63
self-esteem (RSES)	35.67	4.28	35.23	4.60	35.76	4.35	33.39	4.43	33.86	4.44
self-efficacy (SWE)	29.97	4.72	31.00	3.96	30.24	4.38	30.23	4.52	30.32	4.68
emotion regulation (SEK-27)	83.13	11.76	82.90	13.11	75.96	14.80	73.87	16.47	74.43	16.05
resilience (CD-RISC)	72.03	11.89	73.80	12.75	71.12	12.23	69.81	13.57	69.43	13.16
active coping (Brief-COPE)	5.73	1.48	5.43	1.55	5.24	1.42	5.06	1.18	5.07	1.21
positive reframing (Brief-COPE)	5.30	1.42	5.60	1.28	5.52	1.53	4.90	1.11	4.82	1.44
stressor load (MIMI; occurrence)	59.20	24.44	55.43	27.25	62.28	31.97	68.61	37.00	58.44	26.72

**Table 4 ijerph-19-00649-t004:** Results of group differences over the follow-up period (t1–t4) using linear mixed model analysis adjusted for the outcome at t0.

Outcome Measure	β	t	*p*	*η* ^2^
mental health (GHQ-28)	−4.18	−2.67	0.01	0.11
wellbeing (WHO-5)	2.62	2.97	0.004	0.13
resilience (BRS)	0.07	0.66	0.51	0.01
satisfaction with life (SWLS)	1.69	1.82	0.07	0.05
perceived stress (PSS-10)	−0.84	−1.41	0.16	0.03
self-esteem (RSES)	0.43	0.81	0.42	0.01
self-efficacy (SWE)	0.84	1.49	0.14	0.03
emotion regulation (SEK-27)	5.41	2.44	0.02	0.09
resilience (CD-RISC)	3.46	2.14	0.04	0.07
active coping (Brief-COPE)	0.64	2.74	0.01	0.11
positive reframing (Brief-COPE)	0.59	2.61	0.01	0.10

## Data Availability

The data sets generated and analysed during the current study are available from the corresponding author on reasonable request.

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
