# Peer review of "A Group Intervention to Promote Resilience in Nursing Professionals: A Randomised Controlled Trial"

_ijerph, 2022, doi:10.3390/ijerph19020649_

Round 1
Reviewer 1 Report
Thank you for the opportunity to review this important work. Please find a few suggestions below for improving the manuscript by section.
Abstract
- Sentences should not begin with numbers please revise line 14 to ensure the sentence starting with 72 nurses is modified to not start with a numeral.
Introduction
- The authors provided a nice description of the six psychological resilience factors that were used to develop the intervention
- Clarifying the aim of the study to include the name of the intervention or rearranging the section so that the description of the intervention appears prior to the aim of the study would improve the clarity of this section.
Procedure and recruitment
- Additional detail about the recruitment site (size, number of staff, etc.) would be helpful to help the reader make a determination about generalizability.
Outcome measures
- There is no mention of whether the authors were granted permission to utilize instruments and/or these instruments were part of public domain.
Intervention
- The intervention is described well however it is unclear the process for developing the intervention which should be included.
Statistical analysis
- There is no mention of how data missingness was evaluated and or addressed. It is highly unlikely that there was no data missing in the sample especially given the repeated measures design.
- It is not clear whether the analysis was completed using originally assigned groups or if data for those that dropped out of the study was removed prior to analysis.
Results
- Was the sample representative of the study site? This is important information to share with the reader.
Discussion
- This is not a pilot study it is an RCt and should not be referred to as a pilot study. In the conclusion the authors reference testing the intervention for efficacy which does not occur in pilot studies.
Limitations
- There is not discussion of generalisability or lack of generalisability
General comments
- Because this is a randomised control trial reference to the CONSORT statement http://www.consort-statement.org/ should occur and all items should be reported on as directed by the CONSORT checklist.
- Were all registered nurses included (those working in direct patient care as well as those in leadership roles)? This should be clarified for the reader
Author Response
Reviewer #1
Comment 1: Abstract – Sentences should not begin with numbers please revise line 14 to ensure the sentence starting with 72 nurses is modified to not start with a numeral.
Response 1: This sentence was changed on page 1 line 14: “In total, 72 nurses were recruited and randomised to either an intervention condition or to a waiting list control condition.”
Comment 2: Introduction – The authors provided a nice description of the six psychological resilience factors that were used to develop the intervention
Clarifying the aim of the study to include the name of the intervention or rearranging the section so that the description of the intervention appears prior to the aim of the study would improve the clarity of this section.
Response 2: This information was added on page 2 line 49-51: “The studied group intervention (which we called “The new Growth”) was designed to promote nurses’ mental health and resilience by helping them to better cope with their work-related and private stressors.”
Comment 3: Procedure and recruitment – Additional detail about the recruitment site (size, number of staff, etc.) would be helpful to help the reader make a determination about generalizability.
Response 3: This information was added on page 4 line 187-189: “To the time of the study, the University Medical Centre had approximately 8,500 employees including about 4,000 nurses, and approx. 1,600inpatient beds.”
Comment 4: Outcome measures – There is no mention of whether the authors were granted permission to utilize instruments and/or these instruments were part of public domain.
Response 4: Licence to use the respective tests is regulated by the statutes of the Leibniz-Insitute for Resilience Research.
Comment 5: Intervention – The intervention is described well however it is unclear the process for developing the intervention which should be included.
Response 5: We agree with the reviewer that a more detailed description of the development would be useful. We added this information on page 5 line 239-263: “The intervention was developed with the aim of providing participants with new skills to help them cope better with individual stressors. The intervention based on a psychodynamic model about the development of stress. In the first two training sessions, the nurses' private and occupational problems were analysed and an initial psychodynamic understanding of the stressors was developed. The nurses considered how the experience of stress relates to difficult relationship experiences in their childhood. Building on this, they learned which inner psychic conflicts are the triggers for their stress. They also identified frequently used dysfunctional defence mechanisms (e.g. rationalisation, regression, reaction formation, isolation of affect, autoaggression, etc.). This psychodynamic model formed an important basis for the rest of the training and was always an indirect part of the other sessions as well. For the development of sessions three to five, six resilience factors were initially selected based on their empirical evidence: cognitive flexibility, coping, self-efficacy, self-esteem, self-care and mindfulness. In order to improve these factors, exercises based on various therapeutic techniques were developed. For example, mentalization exercises and exercises about cognitive restructuring to train cognitive flexibility, PMR and mindfulness exercises to improve self-care or mindfulness, gathering strengths and skills to improve self-efficacy, etc. Training sessions six and seven comprised a problem-solving process in which the nurses focused on one of their main problems from the beginning of the training. In order to solve this problem, the nurses selected their appropriate strategies and skills, transferred them to everyday life and tested them for their usefulness. The last training session included reflections on personal development over the past eight weeks. There were exercises for a more realistic and positive self-image in which the participants gave each other feedback. In addition, discussions about one's own values in life and the goals after the training were initiated. Exercises were done in form of individual, pair and group work. Homework assignments were incorporated to consolidate the subject matter. The training was tested for feasibility in a preliminary study and then slightly modified again.”
Comment 6: Statistical analysis – There is no mention of how data missingness was evaluated and or addressed. It is highly unlikely that there was no data missing in the sample especially given the repeated measures design.
It is not clear whether the analysis was completed using originally assigned groups or if data for those that dropped out of the study was removed prior to analysis.
Results
Was the sample representative of the study site? This is important information to share with the reader.
Response 6: We agree with the reviewer’s comments that more details are needed here. Subjects with missing values were excluded from the analysis using ANCOVA. Subjects with missing data were not excluded from the analyses using linear mixed models (LMM). LMMs used all available observations from participants with some observations missing, but missing observations were not imputed. We added this Comment on page 6 line 284-286: “Subjects with missing values were excluded from the analysis using ANCOVA, but not from linear mixed models. Missing data was not imputed.”
We also added the following sentence on page 7 line 314-316: “With regard to the entire nursing staff of the clinic, the representativeness of the sample can be assumed, since nurses from many different medical departments participated in the study.”
Comment 7: Discussion – This is not a pilot study it is an RCt and should not be referred to as a pilot study. In the conclusion the authors reference testing the intervention for efficacy which does not occur in pilot studies.
Response 7: We agree with the reviewer and changed the designation in the abstract on page 1 line 13 and in the discussion on page 10 line 357.
Comment 8: Limitations – There is not discussion of generalisability or lack of generalisability
Response 8: We followed the reviewer´s recommendation and added this Comment in the Limitations on page 12 line 435-437: “Although the sample consisted of nurses from various medical departments, it cannot be clearly determined whether the results are representative of other hospitals or other cultural circumstances.”
and on page 12 line 442-443: [A multicentre study in several hospitals could be considered in order to achieve a larger sample size and reduce clinic-specific influencing factors.] “This would further improve the representativeness of the sample and the generalisability of the results.”
Comment 9: General comments – Because this is a randomised control trial reference to the CONSORT statement http://www.consort-statement.org/ should occur and all items should be reported on as directed by the CONSORT checklist.
Were all registered nurses included (those working in direct patient care as well as those in leadership roles)? This should be clarified for the reader
Response 9: We filled in the CONSORT checklist and propose to add the checklist in the appendix. We also added the following information on page 4 line 183-184: “In the appendix, the CONSORT checklist provides information on fulfilment of CONSORT requirements.”
We also added the following sentence on page 4 line 191-192: “Nurses from direct patient care and nurses in leadership positions were recruited for the study.”

Reviewer 2 Report
The manuscript entitled "A group intervention to promote resilience in nursing professionals: A randomized controlled trial" is an excellent example of a well-designed and reported study that provides evidence of the efficacy of a novel therapeutic intervention program for nurses. Although the number of participants is not large, the prospective design of repeated measures using randomized controlled trials allows conclusions to be drawn from the study. The introduction is comprehensive, the methods are well described, and the results are greatly presented in tables and figures (which improves clarity and transparency). Moreover, no ethical issues were identified. Moreover, the value of the study increases during the COVID-19 pandemic, as nurses need comprehensive prevention and intervention programs to support mental health in these challenging times. As such, this article is worthwhile and timely.
Author Response
Reviewer #2
Comment 1: The manuscript entitled "A group intervention to promote resilience in nursing professionals: A randomized controlled trial" is an excellent example of a well-designed and reported study that provides evidence of the efficacy of a novel therapeutic intervention program for nurses. Although the number of participants is not large, the prospective design of repeated measures using randomized controlled trials allows conclusions to be drawn from the study. The introduction is comprehensive, the methods are well described, and the results are greatly presented in tables and figures (which improves clarity and transparency). Moreover, no ethical issues were identified. Moreover, the value of the study increases during the COVID-19 pandemic, as nurses need comprehensive prevention and intervention programs to support mental health in these challenging times. As such, this article is worthwhile and timely.
Response 1: We thank Reviewer 2 for this positive feedback.
Reviewer 3 Report
The manuscript presents interesting findings that contribute to increase the evidence of interventions that may improve health workers capacity to cope with stress in occupational settings. Design and statistical analysis are well conducted and presented.
I would only suggest some minor changes.
1) More theoretical details should be given concerning the integration between cbt and psychodinamic approach in the introduction and in the "intervention" paragraph of methods section, as long with a more datailed overview of the intervention's contents;
2) In the outcome measure paragraph, the term " classified as psychiatric" could be sobstituted with a less medicalized and illness base terminology, such as "presence of mental health symptoms": psychiatry is a discipline, not a diagnosis.
3) In the results section, it could be interesting to report the total number of nurses working at the Medical Centre of the Johannes Gutenberg-University to have a general idea of the impact that the intervention had at an organizational level. More details about the organizational impact of the intervention could be interesting for HR policy makers.
Author Response
Reviewer #3
Comment 1: More theoretical details should be given concerning the integration between cbt and psychodinamic approach in the introduction and in the "intervention" paragraph of methods section, as long with a more datailed overview of the intervention's contents.
Response 1: We agree with the reviewer that a more detailed description of the development would be useful. We added this information on page 2 line 86-92: “Psychotherapy research shows that, depending on the patient and the context, the integration of therapeutic approaches from different schools can effectively contribute to the success of the treatment, is more flexible to patients' needs and allows a better adaptation of the therapy. The studied intervention allows a certain flexibility in the implementation. The nurses can set their own priorities and have free choices between exercises from different therapeutic approaches.”
See also our response to comment 5 of reviewer 1 regarding added information about the intervention's contents (see page 5 line 239-263): “The intervention was developed with the aim of providing participants with new skills to help them cope better with individual stressors. The intervention based on a psychodynamic model about the development of stress. In the first two training sessions, the nurses' private and occupational problems were analysed and an initial psychodynamic understanding of the stressors was developed. The nurses considered how the experience of stress relates to difficult relationship experiences in their childhood. Building on this, they learned which inner psychic conflicts are the triggers for their stress. They also identified frequently used dysfunctional defence mechanisms (e.g. rationalisation, regression, reaction formation, isolation of affect, autoaggression, etc.). This psychodynamic model formed an important basis for the rest of the training and was always an indirect part of the other sessions as well. For the development of sessions three to five, six resilience factors were initially selected based on their empirical evidence: cognitive flexibility, coping, self-efficacy, self-esteem, self-care and mindfulness. In order to improve these factors, exercises based on various therapeutic techniques were developed. For example, mentalization exercises and exercises about cognitive restructuring to train cognitive flexibility, PMR and mindfulness exercises to improve self-care or mindfulness, gathering strengths and skills to improve self-efficacy, etc. Training sessions six and seven comprised a problem-solving process in which the nurses focused on one of their main problems from the beginning of the training. In order to solve this problem, the nurses selected their appropriate strategies and skills, transferred them to everyday life and tested them for their usefulness. The last training session included reflections on personal development over the past eight weeks. There were exercises for a more realistic and positive self-image in which the participants gave each other feedback. In addition, discussions about one's own values in life and the goals after the training were initiated. Exercises were done in form of individual, pair and group work. Homework assignments were incorporated to consolidate the subject matter. The training was tested for feasibility in a preliminary study and then slightly modified again.”
Comment 2: In the outcome measure paragraph, the term " classified as psychiatric" could be sobstituted with a less medicalized and illness base terminology, such as "presence of mental health symptoms": psychiatry is a discipline, not a diagnosis.
Response 2: We basically agree with reviewer´s opinion. The classification as “psychiatric case” stems from the test authors themselves. That’s why we changed this comment slightly on page 5 line 217-217: “A total score of 23 or below can be classified as a non-psychiatric case, while a total score ≥ 24 may be classified as a probable psychiatric case, but this score is not an absolute cut-off.”
Comment 3: In the results section, it could be interesting to report the total number of nurses working at the Medical Centre of the Johannes Gutenberg-University to have a general idea of the impact that the intervention had at an organizational level. More details about the organizational impact of the intervention could be interesting for HR policy makers.
Response 3: We agree with the reviewer and refer to comment 3 of reviewer 1: We added this Comment on page 4 line 187-189: “To the time of the study, the University Medical Centre had approximately 8,500 employees including about 4,000 nurses, and approx. 1,600 inpatient beds.”
We also added the second Comment on page 12 line 455-462: “The organisational impact of the intervention could be investigated in a further study. Influencing mediator and moderator variables should be determined and the effects on job satisfaction and other work-related factors should be examined. This is particularly important because a firm implementation of the intervention in a hospital would involve financial and time investments. Nevertheless, the positive results suggest that a long-term implementation could potentially lead to higher job satisfaction, less fluctuation and improved quality of care, which could be convincing facts for employers and insurance companies to invest.”